# The Effect of Mental Activation of One’s Pet Dog on Stress Reactivity

**DOI:** 10.3390/ijerph20216995

**Published:** 2023-10-30

**Authors:** Kerri E. Rodriguez, Dan J. Graham, Rachel G. Lucas-Thompson

**Affiliations:** 1Human-Animal Bond in Colorado, School of Social Work, Colorado State University, Fort Collins, CO 80523, USA; kerrirodriguez@arizona.edu; 2Department of Psychology, Colorado State University, Fort Collins, CO 80523, USA; dan.graham@colostate.edu; 3Colorado School of Public Health, Colorado State University, Fort Collins, CO 80523, USA; 4Department of Human Development and Family Studies, Colorado State University, Fort Collins, CO 80523, USA

**Keywords:** stress, human–animal interaction, social support, dogs

## Abstract

Research suggests that mental activation of human social support may reduce stress reactivity. However, the extent to which social support from pets elicits a similar effect has been less explored. This study aims to determine whether the mental activation of one’s pet dog reduces stress reactivity to a subsequent experimental stressor. In a 2 × 2 design, 132 dog-owning participants (M_age_ = 20.14; 80% female) were randomly assigned to one of two mental activation conditions (pet dog; general) and one of two stressor conditions (social-evaluative; cognitive). Data were analyzed with two-way ANOVAs with self-reported (positive/negative affect, negative self-evaluation) and physiological (blood pressure, heart rate) dependent variables. Results indicated that participants randomized to the pet dog mental activation condition had smaller decreases in positive affect from baseline to post-stressor compared to the general mental activation condition. However, there were no significant interactions between time and mental activation condition on negative affect, negative self-evaluation, heart rate, or blood pressure. Thus, the mental activation of one’s pet dog had a minimal effect on stress reactivity to a cognitive or social-evaluative stressor. Results suggest that the physical presence of an animal may be an essential mechanism underlying the benefits of animal-derived social support.

## 1. Introduction

### 1.1. Background and Significance

Social support has long been recognized as vital to human health and wellbeing [1,2]. Research has shown that greater quality and quantity of social relationships are linked to reduced mortality, suggesting that biopsychosocial mechanisms link social support and health [2,3,4]. Early conceptualizations of social support [5] posited that social support is a multi-faceted construct that can be categorized into four main domains: emotional support (the provision of caring, empathy, love, and trust), instrumental support (the provision of tangible goods and services or tangible aid), informational support (communication of information that is relevant to problem solving) and appraisal support (communication of information that is relevant to self-evaluation) [6]. Greater perceived support from these domains has been found to attenuate, or buffer, the psychological and physiological effects of stressful events [7,8]. In this context, it is posited that perceived social support from others can modify appraisals of a stressful situation [8]. This stress-buffering model of social support has been supported by several studies finding that individuals exhibit lower acute stress reactivity during an experimental stressor when accompanied by a supportive friend, romantic partner, or family member compared to when alone [9,10,11,12]. In particular, individuals who undergo experimental stressors with social support not only self-report less stress during and after the stressor but also exhibit attenuated physiological reactivity, particularly in cardiovascular parameters such as blood pressure and heart rate [12,13,14].

In addition to human social support, research has recently begun to examine the effect of animal-derived social support on stress reactivity. Historically, companion animals (i.e., pets) have been recognized as providers of instrumental social support by providing tangible services such as herding, guarding, hunting, etc. However, pets are increasingly recognized for their ability to provide emotional support, both in formal roles (e.g., as therapy animals) and as companion animals. Many pet owners in the US consider their pets members of their family [15] that provide significant emotional support, including during times of stress [16,17]. Specifically, due to animals’ ability to provide perceived non-judgmental positive regard and empathy [18], animals may be particularly beneficial in providing emotional support in similar ways as humans.

Indeed, studies have found that the presence of a friendly dog reduces self-reported and cardiovascular measures of stress reactivity during a stressful procedure compared to when alone [19,20,21] or accompanied by a stuffed animal [22,23]. Other studies have also found that both the presence of one’s own dog or an unfamiliar friendly dog can lead to lower stress reactivity during an experimental stress paradigm than when accompanied by a form of human social support, such as a parent [24], a friendly adult [25,26], or a friend [27]. Because the presence of a human during a challenging task can create evaluation apprehension that may actually exacerbate the stress response, companion animal-derived social support may be more beneficial for reducing stress reactivity due to their non-judgmental nature [27].

Existing studies examining the effect of companion animals on stress reactivity have focused on a single type of stressor. However, to our knowledge, there has not been a study directly comparing how animal-derived social support may differentially impact stress reactivity depending on the nature of the stressor (i.e., a cognitive stressor vs. a social stressor). A meta-analysis in 1999 found that human-derived social support was, overall, related to lower heart rate, systolic blood pressure (SBP), and diastolic blood pressure (DBP) reactivity but found evidence of significant moderators depending on the support provider and the type of stressor (i.e., conducting a speech versus performing arithmetic or anagrams) [9]. As the stress-buffering theory of social support suggests that buffering effects depend on the characteristics of the stressor and the “need” that the supportive relationship can fill [7], the non-judgmental support that an animal provides may be more impactful in the context of a social-evaluative stressor (where one could be negatively judged by others) compared to a cognitive stressor.

### 1.2. Mental Activation of Social Support

Because a supportive figure may not be physically present during all stressful situations that occur in life, of interest is the possibility that simply thinking about a social tie may be beneficial. Specifically, researchers have investigated if mentally activating an internal representation of social support (i.e., support schemas) may also significantly buffer stress reactivity to a stressor. In one of the first studies to investigate this, Smith et al. [28] found evidence that mental activation of social support may be similarly effective in lowering stress reactivity as physically present social support. Specifically, participants that wrote and thought about a supportive tie had lower heart rate and blood pressure reactivity to a subsequent laboratory stressor than those that wrote and thought about a casual acquaintance [28]. Other studies have found that completing a structured writing exercise to activate a support schema before completing a stressor attenuates cardiovascular reactivity compared to no social support priming or mental activation of a casual acquaintance [28,29,30]. In addition to writing tasks, some studies have found that viewing photographs of a social tie can attenuate feelings of physical pain [31] and reduce distress [32]. These findings align with theoretical models of social relationships and attachment, which suggest that individuals develop internal mental representations, or internal working models, of their relationships [33,34]. When these schemas are mentally activated, they can significantly impact, or “prime”, subsequent emotion, motivation, and behavior [35,36]. In the context of a stressor, this social support priming is hypothesized to temporarily alter an individual’s self-concept to reduce subsequent appraisals of threat [28,37].

In comparison to human social support, the effect of mental activation of animal-derived social support on stress reactivity has been much less researched. A seminal 2012 study conducted two experiments to determine if mentally activating an attachment relationship with a pet could provide similar benefits as having a pet physically present. The study found that participants who wrote about their relationship with their pet prior to the stressor (cognitive presence) and participants who had their pet present (physical presence) both had lower blood pressure during the task compared to a control group [38]. Further, participants in both the cognitive and physical presence groups were less likely to appraise a subsequent task as a threat than the control group. Thus, this study found that both physical and mentally-activated support from one’s pet may provide comfort, regulate distress, and potentially act as a secure base [38]. In 2019, another study found that viewing a picture of a pet during a cognitive stressor led to higher subjective ratings of feeling relaxed but had no effect on cardiovascular reactivity compared to viewing pictures of an unfamiliar animal or image of nature [39]. However, this study did not prime participants with their relationship to their pet prior to the stressor, deviating from prior methods. Due to the limited studies in this area and inconsistent findings and methodology, more research is needed to determine the effect of mental activation of animal-derived social support on stress reactivity. Specifically, it remains unclear how, and when, activating a support schema of one’s pet may be beneficial.

### 1.3. Aims of the Current Research

The first aim of the current study was to determine whether mental activation of one’s pet dog reduces stress reactivity during a mild stressor. We hypothesized that individuals who completed a structured writing activity about their pet dog prior to completing a stressor would report higher positive affect and less cardiovascular stress reactivity in response to the stressor than those who completed a structured writing activity about a general positive memory. As a second aim, we assessed the extent to which the effect of this mental activation on stress reactivity may differ based on the type of stressor (social-evaluative or cognitive). 

## 2. Materials and Methods

### 2.1. Participants

Participants were N = 132 dog-owning undergraduate and graduate students over the age of 18 attending a large public university in the United States. We restricted participants to dog owners to provide consistency and because relationships with pet dogs tend to be stronger than relationships with other types of pets [40]. Participants were between 18 and 33 years old (M = 20.14; SD = 2.21) and mostly identified as female (80% female; 20% male). Participants mostly identified as Caucasian (92% Caucasian; 5% American Indian or Alaskan Native; 2% Asian/Pacific Islander; 4% Black or African American; 3% Other; categories were non-exclusive, resulting in percentages adding to more than 100%). Participants were recruited from several sources, including the Psychology Department research pool, emails sent to professors of Psychology and Human Development and Family Studies courses, and flyers posted in buildings in which these classes occurred. Participants recruited from the research pool completed the study for course credit, while the other students were entered to win a $25 Target gift card as remuneration for their participation.

### 2.2. Study Design

All protocols were approved by the Purdue University Institutional Review Board (Protocol #13-4660H). Participants were randomized to one of two mental activation conditions (asked to think/write about a memory of their pet dog or asked to think/write about any positive memory as a comparison) and to one of two stressor conditions (social-evaluative stressor or cognitive stressor). Randomization occurred via a random number generator (random.org). Participants were blinded to their assignment. Thus, all participants were asked to bring a digital or physical photo of their pet dog to the laboratory session regardless of which mental activation condition they were allocated. The research team was not blinded to participant assignment to conduct the correct protocols during each participant’s visit, which varied by assignment. The analyzed sample consisted of four groups: N = 37 cognitive stressor/pet dog mental activation group (Cog_Dog_); N = 28 cognitive stressor/general positive memory mental activation group (Cog_Gen_); N = 35 social-evaluative stressor/pet dog mental activation group (SE_Dog_); and N = 32 social-evaluative stressor/general positive memory mental activation group (SE_Gen_).

### 2.3. Procedures

All procedures occurred during a single visit to an observational laboratory over 35 min (Figure 1). After obtaining informed consent, a research assistant attached a blood pressure cuff to the participant’s arm to collect baseline BP and HR. During a 10-min baseline phase, participants sat quietly and looked through a provided nature magazine. At the end of the 10-min, participants completed a self-report survey to obtain baseline affect and self-evaluation. Next, a 10-min mental activation phase began. Participants were randomized to either a pet dog mental activation or a general positive memory mental activation. Those randomized to the pet dog condition were asked to think about their dog, write a memory of their dog, and then read to the researcher what they wrote aloud, replicating methods by previous studies replicating methods by previous studies [30,41]. To further strengthen the activation of the support schema, participants also viewed a picture of their dog during this period. Participants randomized to the general positive memory condition were asked to write about a positive memory and similarly read to the researcher what they wrote. The research team conducted a post hoc review of the written memory text to ensure fidelity in each condition; reviews indicated that all memories from the dog condition contained a discussion of the pet dog and that no participants assigned to the general positive memory condition wrote about a memory with their pet dog. 

After the mental activation phase, the stressor phase began. Participants were randomized to either a social-evaluative stressor or a cognitive stressor. Both stressors lasted approximately 15-min, and physiological data were collected every 3 min. At the end of the stressor, participants completed another self-report survey to indicate post-stressor affect.

Participants randomized to the social-evaluative stressor were administered the Trier Social Stress Test (TSST) [42]. The TSST is a public performance test that has been shown to reliably induce mild-to-moderate stress and accompanying physiological responses in most participants [43]. The first phase of the TSST consists of a 5-min preparation phase in which participants are told to prepare a speech as part of a mock job interview. Participants were told that an evaluator with special training in analyzing behavior would observe the speech and that the speech would be videotaped to be later analyzed by experts. In the 5-min speech phase, participants were asked to deliver the speech in front of an evaluator who, unbeknownst to participants, was trained to remain neutral during the task. Participants then completed a 5-min mental arithmetic phase in which they were asked to continuously subtract seven from 1027 aloud in front of the evaluator without making mistakes and as quickly as possible until they reached zero. If a participant made a mistake, they were stopped by the evaluator and told to restart from the beginning. At the end of the mental arithmetic phase, the TSST ended, and the research team thoroughly debriefed the participant; the researchers explained that participants were not actually being evaluated or recorded and that the ”evaluator” was just a confederate of the research team who was trained to remain neutral during the task. If at any point the participant exhibited severe distress or verbally indicated that they would not like to continue, the procedure ended.

Participants randomized to the cognitive stressor completed a mental arithmetic task and a solvable anagram task. During the 12-min mental arithmetic task, participants engaged in a series of subtractions (e.g., subtract seven from 6828) and were prompted to speed up their responses at the start of minutes three, seven, and eleven [44]. If a participant made a mistake, they were corrected by the researcher and asked to continue from the correct number. During minutes 7–12, participants were presented with a series of random 100 db white noise blasts administered via headphones and were told that the noises were intended to make the task more challenging. Afterward, participants conducted a 3 min solvable anagram task [45] in which they were presented with a series of 5-letter anagrams with only one solution. Each anagram was presented on a computer screen and participants were given approximately eight seconds to respond verbally with the correct answer. Although both the cognitive stressor and the social-evaluative stressor involved mental arithmetic, a key element of the TSST is social evaluation [46] which was not intended in the cognitive stressor.

### 2.4. Measures

Demographic information collected from participants included age, gender identity, and race/ethnicity. In addition, participants reported on health behaviors that are associated with physiological functioning (consumption of caffeine, alcohol, and nicotine over the 24 h prior to the visit, medication taken and physical activity the day of the visit, and hours slept the previous night).

Participants reported on their emotional state via the 20-item Positive and Negative Affect Schedule (PANAS) [47] before and after the stressor. The PANAS is one of the most commonly used measures to capture participants’ subjective state during the TSST due to its sensitivity to change and ease of use [48]. Participants were asked the degree to which they felt a series of ten positive emotions (e.g., proud, excited) and ten negative emotions (e.g., scared, upset) in the current moment on a five-point Likert scale (1 = “not at all” to 5 = “extremely”). Scores were calculated by averaging positive items to create a positive affect (PA) score and averaging negative items to create a negative affect (NA) score. The PANAS had good reliability in this sample both at baseline (Cronbach’s α = 0.786 for NA, α = 0.836 for PA) and post-stressor (Cronbach’s α = 0.889 for NA, α = 0.889 for PA). 

Because expert guidelines suggest a minimum of two different self-report measures to assess participants’ subjective stress state [48], participants were also asked to rate the extent to which they felt a set of ten emotions related to feelings of negative self-evaluation before and after the stressor (e.g., self-conscious, embarrassed, defeated) on a five-point Likert scale (1 = “not at all” to 5 = “extremely”). Scores were calculated by averaging items to create an overall negative self-evaluation (NSE) score. This measure had good reliability in this sample (Cronbach’s α = 0.836 for baseline NSE, α = 0.938 for post-stressor NSE). 

Objective stress reactivity was measured via systolic and diastolic blood pressure (SBP, DBP) and heart rate (HR) to accompany subjective measures. A noninvasive fitted cuff monitor (Dinamap Pro 100V2, GE Medical Systems, Tampa, FL, USA) was used to measure BP and HR, replicating protocols from previous studies [49,50]. The cuff monitor was affixed to the participant’s nondominant arm and collected measurements automatically every 3 min in the seated position throughout the procedure. Average SBP, DBP, and HR values were then calculated for each phase (baseline, mental activation, and stressor).

### 2.5. Analytic Plan

All analyses were conducted in SPSS version 28.0 (IBM Statistics, Armonk, NY, USA). One-way ANOVA and chi-squared tests were used to ensure that randomization resulted in no significant between-group differences in demographic continuous and categorical variables. SBP, DBP, and HR data were screened for outliers, and outliers over three standard deviations above or below the mean were winsorized (*n* = 8 data points). Kolmogorov–Smirnov tests were run to examine the normality of variables, and DBP, PA, NA, and NSE were log-transformed for analyses. Statistical analyses were performed with the log-transformed data, but tables and figures report the raw data. To analyze the effect of experimental conditions on outcomes, two-way repeated-measures ANOVA tests were conducted on each outcome with a within-subjects variable of time (baseline, post-stressor for PA, NA, and NSE; baseline, mental activation, and stressor for SBP, DBP, and HR) and between-subjects independent variables of mental activation conditions (pet dog or general positive memory) and stressor conditions (social-evaluative vs. cognitive). Therefore, time×condition interactions were used to examine differences between memory and stressor conditions over time. For all models, sphericity was determined with Mauchly’s test of sphericity, and Greenhouse–Geisser (if ε < 0.75) or Huynh–Feldt (if ε ≥ 0.75) corrections were performed as necessary. The significance level was set at *p * <  0.05, and effect sizes were interpreted at η^2^ = 0.01 as a small effect, η^2^ = 0.06 as a moderate effect, and η^2^ = 0.14 as a large effect. To account for inflated Type I errors due to multiple comparisons, we used the false discovery rate (fdr) to adjust all alphas [51]. A repeated measures, within-between interaction post hoc power analysis conducted with G*Power (version 3.1.9.7; Kiel University, Kiel, Germany) indicated that the sample of N = 132 was adequately powered at 0.93 to detect a medium effect (f = 0.15) with 2 repeated measurements (PA, NA, and NSE variables) and at 0.97 for 3 repeated measurements (SBP, DBP, and HR variables).

## 3. Results

Participants did not significantly differ in age (*F*(3,128) = 2.273, *p* = 0.083), gender identity (χ^2^ = 2.103, *p* = 0.551), or race/ethnicity (χ^2^ = 0.445, *p* = 0.931) across experimental groups. In addition, groups did not differ in terms of factors known to affect stress reactivity (i.e., the number of cups of caffeine, cigarettes, or alcoholic drinks ingested on the day of testing nor the number of hours slept the night before nor exercise presence and intensity on the testing day; *p*s > 0.514). Therefore, these variables were not considered further in analyses. 

### 3.1. Self-Reported Outcomes

Table 1 displays the descriptive statistics and statistical analysis output of positive affect (PA), negative affect (NA), and negative self-evaluation (NSE) by condition, while Figure 2 displays the descriptive statistics graphically. Results indicated that for all three self-reported outcomes, there was a significant within-subject effect of time such that there was a significant decrease in positive affect, a significant increase in negative affect, and a significant increase in negative self-evaluation (all significant after fdr adjustment) from baseline to after the stressor. These changes were all large in effect size. 

For positive affect, there was a significant time×memory condition effect with a small-to-moderate effect size. A follow-up independent *t*-test on the change score for positive affect found that participants in the general positive memory group had significantly larger decreases in their positive affect from baseline to after the stressor compared to the pet dog memory group with a medium effect size (*t*(130) = 2.276, *p* = 0.024, Cohen’s *d* = 0.62, significant after fdr adjustment). There was no significant time×stressor condition effect and no significant time×stressor×memory effect for positive affect.

In contrast, the increase in negative affect and negative self-evaluation from baseline to post-stressor did not significantly differ by memory condition or type of stressor received. For negative affect, there were no significant time×memory condition, time×stressor condition, or time×stressor×memory condition effects (all with negligible effect sizes). Similarly, for negative self-evaluation, there were no significant time×memory condition, time×stressor condition, or time×stressor×memory condition effects (all with negligible effect sizes).

### 3.2. Physiology Outcomes

Table 2 displays the descriptive statistics and statistical analysis output for the physiological outcomes of systolic blood pressure (SBP), diastolic blood pressure (DBP), and heart rate (HR) by condition. Figure 3 displays the descriptive statistics graphically. Results indicated a significant within-subjects effect of time on SBP, DBP, and HR (all significant after fdr adjustment). These changes were all large in effect size. Specifically, participants exhibited significantly higher SBP, DBP, and HR during the stressor than during the baseline condition (*p*s < 0.001) and during the mental activation period (*p*s < 0.001). There were no significant differences in SBP and DBP between the baseline and mental activation periods (*p*s > 0.057). However, HR was higher in the mental activation period than in the baseline period (*p* < 0.001). 

For SBP, there was no significant time×memory condition effect. However, there was a significant time×stressor condition effect (significant after fdr adjustment) that was small to moderate in size. A follow-up independent *t*-test on the change score for SBP from pre- to post-stressor found that participants in the social-evaluative stressor group had a significantly larger increase in SBP compared to the cognitive stressor group with a medium effect size (*t*(130) = 3.353, *p* = 0.001, Cohen’s *d* = 0.58, significant after fdr adjustment; Figure 3). Finally, there was no significant time×stressor×memory condition effect. 

For DBP and HR, there was no significant time×memory condition, time×stressor condition, or time×stressor×memory condition effect. Therefore, the increase in DBP and HR from baseline to stressor did not significantly differ by memory condition or type of stressor received. 

## 4. Discussion

The objective of this study was to determine to what extent mental activation of one’s pet dog reduces stress reactivity to a social-evaluative or cognitive stressor. Results suggested that participants who wrote and thought about their pet dog prior to an experimental stressor reported less of a decrease in positive affect from before to after the stressor compared to participants who wrote and thought about a general positive memory. However, this study found that mental activation conditions did not differ in the magnitude of change in other subjective stress measures (negative affect, negative self-evaluation) or physiological stress (SBP, DBP, HR). The impact of mental activation of a pet dog on stress reactivity was also not dependent on the type of stressor received (social-evaluative or cognitive). 

Findings from the current study differ from those of Zilcha-Mano and colleagues, who found that mental activation of one’s pet dog reduced subsequent cardiovascular reactivity during a stressor [52]. Importantly, the studies differed in the population (students from the US vs. adults from Israel), the type of stressor used (TSST/anagram task vs. Remote Associates Test), and the environment (laboratory vs. home setting), so it is difficult to make direct comparisons. However, findings do partially replicate those of Ein and colleagues, who found that pet-owning adults randomized to view a picture of their pet did not have significantly different cardiovascular reactivity to the TSST in comparison to those who viewed a picture of an unfamiliar pet, a familiar person, an unfamiliar person, an image of nature, or no image at all [39]. Due to methodological differences across studies, more research is needed to add to this growing literature base to explore the role of mental activation of pets in stress reduction.

Previous studies have found that mentally activating internal representations of a supportive human relationship can attenuate physiological stress reactivity during a stressor [28,29,30]. However, these studies specifically asked participants to think of and write about a relationship they perceived as supportive in their life. Therefore, no manipulation check is required to ensure that the relationship elicits feelings of social support, which is the proposed mechanism by which this psychophysiological attenuation is thought to occur [7]. In contrast, this study involved mental activation of a relationship with one’s pet dog, which could have ranged widely in perceived supportiveness. Specifically, individuals who view their pet dog as more emotionally supportive in their lives may benefit more from mentally activating their pet dog before the stressor than individuals who view their pet dog as less emotionally supportive. Future research is needed to understand the role of emotional support and the strength of the human–animal bond may relate to outcomes. Future studies could specifically recruit participants who self-report high feelings of emotional support from their dogs to limit this variability or could examine how variability in pet-derived emotional support may be related to subjective and objective measures of stress reactivity in a general population,

Results lend evidence towards the possibility that the physical presence of a dog may be required to elicit a significant stress-reducing effect. Indeed, several studies have found significant stress-buffering effects of dog presence via reduced heart rate [19,23], electrodermal activity [21], cortisol [27], and blood pressure [22] in response to a stressor compared to when alone or with a supportive human relationship. However, other studies have found that dog presence only impacts self-report measures (e.g., positive affect, state anxiety), but not physiological measures [24,53,54]. Differences in results have been attributed to the possibility that physical touch, such as petting or stroking, may be a key mechanism underlying these effects. For example, studies have found that the more time participants spend stroking a friendly dog during the TSST stressor, the larger the decrease in the stress hormone cortisol [25]. Other studies have found that individuals who physically interact with a dog report greater improvements in wellbeing compared to those in proximity to a dog without engaging in physical contact [41] or those that viewed pictures of a dog [52]. Future research is necessary to explore the degree of physical proximity and contact required with a dog to achieve a stress-buffering effect compared to mental activation alone.

Our second hypothesis, which stated that the stress-buffering effect of thinking about one’s pet dog would be more pronounced among individuals randomized to completing a social-evaluative stressor than those completing a cognitive stressor, was not supported. While we did find that participants who received the social-evaluative stressor had significantly larger increases in SBP compared to the cognitive stressor, there were no significant interactions between time, stressor condition, and memory condition. Therefore, the two stressors resulted in similar increases in cardiovascular stress reactivity regardless of the type of memory condition. It is important to note that although the TSST is described as a social stressor in the literature, it contains cognitive (i.e., mental arithmetic) components completed in a social-evaluative manner. In addition, the cognitive stressor may have contained a small social component given that the experimenter was required to correct the participant in response to an error. This overlap in the nature of the stressors may explain why an effect of stressor type was not found. It is possible that if we had chosen a purely social stressor task such as social exclusion, the pet dog condition may have been more impactful, e.g., [19]. Another important consideration is that this study did not measure hypothalamic–pituitary–adrenal axis (HPA axis) activity. It is possible that we may have found significant differences in stress reactivity via the HPA axis (e.g., cortisol), given that social-evaluative stressors are particularly effective at eliciting HPA axis activity [46].

### Limitations and Future Directions 

Results of this study should be interpreted with the following limitations in mind. First, the sample was not representative; using non-probability convenience sampling, we obtained a sample that was 92% White, 80% female, and all dog-owning undergraduate or graduate students at a university. It is likely that there was also self-selection bias present, such that participants who volunteered for the study likely had a positive relationship with their pet dog. Therefore, findings are not generalizable to the larger population of dog owners. Second, group size was uneven due to limitations to randomization to four groups with a smaller sample. However, because these differences were not due to bias or non-random factors, implications for validity of the study’s findings are likely minimal. 

Finally, as pointed out by previous studies [30], mental activation of a support schema cannot be observed directly and, thus, has unmeasured variation in the quantity and quality of support activated. Participants were not given instructions on what *type* of positive memory to write about; therefore, written memories varied widely and could have prompted different feelings and emotions. Future research will benefit from standardizing the context and type of mental activation across participants in both dog-related and general positive memory conditions. To equalize participant expectations, participants were told to bring (or have access to) a physical or digital picture of their dog when they arrived at the experimental session. Therefore, this requirement could have unintentionally primed participants to think of their pet dog, even if they were randomized to the general positive memory condition. While it is likely that this effect would be short-lived, future research may consider carefully safeguarding against potential priming or expectancy biases. 

## 5. Conclusions

This study found that pet dog owners that mentally activated a memory with their pet dog reported less of a decrease in positive affect from before to after an experimental stressor compared to participants that mentally activated a general positive memory not featuring their dog, regardless of the type of stressor received. However, groups did not significantly differ in cardiovascular stress reactivity, including blood pressure and heart rate, nor in negative affect or negative self-evaluation. Second, the effect of mental activation of one’s pet dog did not significantly differ among those completing a social-evaluative stressor compared to those completing a cognitive stressor. Results suggest that mental activation of one’s pet dog may not be sufficient to elicit a social support schema to significantly impact cardiovascular stress reactivity. Future research is necessary to parse out the level at which individual differences and human–animal interaction characteristics may impact results.

## Figures and Tables

**Figure 1 ijerph-20-06995-f001:**
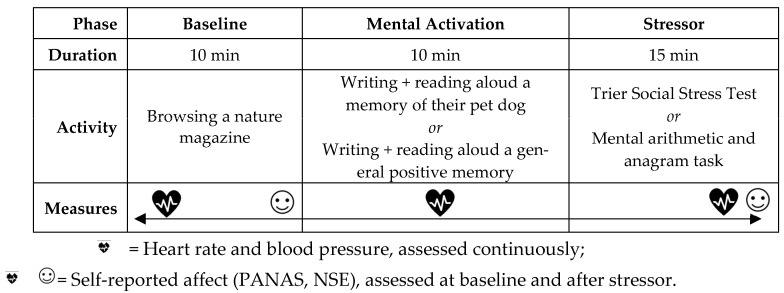
Experimental procedure.

**Figure 2 ijerph-20-06995-f002:**
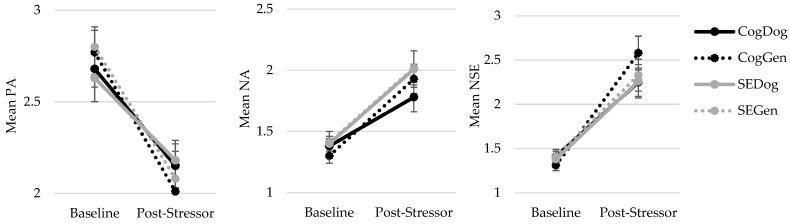
Graphs displaying changes in positive affect (PA), negative affect (NA), and negative self-evaluation (NSE) from baseline to post-stressor across groups.

**Figure 3 ijerph-20-06995-f003:**
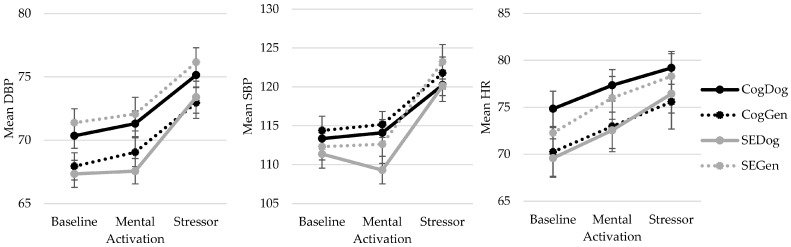
Graphs displaying diastolic blood pressure (DBP), systolic blood pressure (SBP), and heart rate (HR) during the baseline, mental activation, and stressor periods across groups.

**Table 1 ijerph-20-06995-t001:** Descriptive statistics and statistical analyses output for self-reported measures across conditions.

	Condition	Baseline M (SD)	Post-Stressor M (SD)	Time	Time × Memory	Time × Stressor	Time × Memory × Stressor
Positive Affect	Cog_Dog_	2.68 (0.63)	2.15 (0.72)	***F* = 128.152 ***** **η^2^ = 0.500**	***F* = 6.922 **** **η^2^ = 0.051**	*F* = 0.452 η^2^ = 0.004	*F* = 0.179 η^2^ = 0.001
	Cog_Gen_	2.77 (0.63)	2.01 (0.78)				
	SE_Dog_	2.63 (0.74)	2.18 (0.64)				
	SE_Gen_	2.80 (0.64)	2.08 (0.85)				
Negative Affect	Cog_Dog_	1.38 (0.35)	1.78 (0.73)	***F* = 90.941 ***** **η^2^ = 0.415**	*F* = 1.844 η^2^ = 0.014	*F* = 0.144 η^2^ = 0.001	*F* = 0.800 η^2^ = 0.006
	Cog_Gen_	1.30 (0.32)	1.93 (0.66)				
	SE_Dog_	1.40 (0.38)	2.01 (0.91)				
	SE_Gen_	1.41 (0.49)	2.02 (0.79)				
Negative Self-Evaluation	Cog_Dog_	1.41 (0.34)	2.25 (0.98)	***F* = 176.464 ******* **η^2^ = 0.580**	*F* = 2.853 η^2^ = 0.022	*F* = 1.529 η^2^ = 0.012	*F* = 1.625 η^2^ = 0.013
	Cog_Gen_	1.31 (0.31)	2.58 (0.99)				
	SE_Dog_	1.38 (0.35)	2.26 (1.12)				
	SE_Gen_	1.41 (0.44)	2.33 (1.03)				

Note: Cog_Dog_ = Pet dog mental activation and cognitive stressor; Cog_Gen_ = General positive memory mental activation and cognitive stressor; SE_Dog_ = Pet dog mental activation and social-evaluative stressor; SE_Gen_ = General positive memory mental activation and social-evaluative stressor; Bolded values indicate significance; *** = *p* < 0.001, ** = *p* < 0.01.

**Table 2 ijerph-20-06995-t002:** Descriptive statistics and statistical analyses output for physiological measures across conditions.

	Condition	Baseline M (SD)	Mental Activation M (SD)	Stressor M (SD)	Time	Time × Memory	Time × Stressor	Time × Memory × Stressor
Systolic Blood Pressure	Cog_Dog_	113.36 (8.18)	114.10 (10.28)	120.24 (8.29)	***F* = 132.766 ***** **η^2^ = 0.509**	*F* = 0.729 η^2^ = 0.006	***F* = 6.583 **** **η^2^ = 0.049**	*F* = 0.493 η^2^ = 0.004
	Cog_Gen_	114.39 (9.68)	115.16 (8.89)	121.78 (10.89)				
	SE_Dog_	111.36 (10.70)	109.30 (10.48)	120.14 (11.91)				
	SE_Gen_	112.32 (9.78)	112.63 (13.87)	123.23 (12.52)				
Diastolic Blood Pressure	Cog_Dog_	70.36 (6.04)	71.31 (6.21)	75.14 (6.07)	***F* = 101.355 ***** **η^2^ = 0.442**	*F* = 0.583 η^2^ = 0.005	*F* = 0.997 η^2^ = 0.008	*F* = 0.960 η^2^ = 0.00
	Cog_Gen_	67.94 (5.68)	69.05 (6.00)	72.96 (6.58)				
	SE_Dog_	67.34 (6.34)	67.55 (5.93)	73.40 (7.47)				
	SE_Gen_	71.40 (6.23)	72.06 (7.53)	76.15 (6.47)				
Heart Rate	Cog_Dog_	74.85 (11.39)	77.35 (10.19)	79.20 (10.67)	***F* = 74.496 ***** **η^2^ = 0.368**	*F* = 0.169, η^2^ = 0.001	*F* = 1.496 η^2^ = 0.012	*F* = 0.863 η^2^ = 0.007
	Cog_Gen_	70.26 (13.69)	72.98 (14.29)	75.58 (15.29)				
	SE_Dog_	69.60 (12.08)	72.55 (11.46)	76.43 (12.03)				
	SE_Gen_	72.28 (13.07)	76.00 (12.91)	78.31 (13.56)				

Note: Cog_Dog_ = Pet dog mental activation and cognitive stressor; Cog_Gen_ = General positive memory mental activation and cognitive stressor; SE_Dog_ = Pet dog mental activation and social-evaluative stressor; SE_Gen_ = General positive memory mental activation and social-evaluative stressor; Bolded values indicate significance; *** = *p* < 0.001, ** = *p* < 0.01.

## Data Availability

Data are available on request.

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
