# Peer review of "The Effect of Mental Activation of One’s Pet Dog on Stress Reactivity"

_ijerph, 2023, doi:10.3390/ijerph20216995_

Round 1

Reviewer 1 Report (Previous Reviewer 2)

Comments and Suggestions for Authors

The concerns I noted in my initial review have been thoroughly addressed!  This articles makes a substantive and unique contribution to HAI benefit research.  Well-done

Author Response

Thank you! 

Reviewer 2 Report (New Reviewer)

Comments and Suggestions for Authors

Rodriguez and colleagues conducted an experimental study on the effects of  mental activation of one’s pet dog on stress  reactivity in a social-evaluative and a cognitive stressor. They found that mental activation of one’s pet dog (as compared to a control condition) lead to a reduced decline in positive affect. However, contrary to their hypothesis, they did not find an effect for other subjective or physiological assessments. Furthermore, they did not find a more pronouced effect of mental activation of one’s pet dog on subjective and phyisological stress measures for the social evaluative stressor.

This is a well-written, easy to follow Manuscript with an overall sound methodology and a timely research question. There are some short-commings in the methodology of the study, but the authors do a good job discussing these and thus stimulate further research. However, I have some comments that need to be addressed:

1.       Methods section: The number of participants in the 4 different groups differ substantially. It is not clear to me how this happened. Different number of drop outs?  How were drop out defined? The authors should specify this and mention it in the Discussion section.

2.       Methods/Results: In line with this, an a priori sample size calculation is missing. Since there is  abundant research on  the effects of  pet’s  and social support on human stress responses, it should hav be possible to estimate an effect size and perform a sample size caculation.  The authors should at least report a post hoc power analysis for their non-significant findings.

3.       Methods:  The assessment of blood pressure with dinamap is state of the art. However, the usage of dinamap to assess heart rate seems uncommon to me and also limits the interpretability of the heart rate data , which should be consistently measured fo at least some minutes. Also, it is not possible to derive any HRV data from these measures (which might be really interesting from a stress reduction/parasympathetic point of view.  This limitation should be included in the Discussion section.  

4.       Methods/Discussion: Social evaluation in the cognitive stressor The authors state correctly that the TSST also has a cognitive component and might not have been exclusively social evaluative.  However, I did not really understand whether the cognitive stressor also had a social evaluative component? The authors write that: If a participant  made a mistake, they were corrected and asked to continue from the correct number.

Was this correction performed by the experimenter? Or automatically by a computer? If the correction was performed by the experimenter the two stressors both have cognitive and social-evaluative components integrated, which may have caused the failure to find an effect of the stressor type. It would be great if the authors could specify this in the Methods section and – if necessary – also integrate this point in the Discussion section.

5.       Discussion section:  There are at least some other studies that I am aware of, which also did not find an effect of social support from a therapy dog on physiological stress indices, but only on subjective parameters (PANAS, STAI-S).  It might be helpful to include these in the Discussion section, e.g.,

Lass-Hennemann, J., Peyk, P., Streb, M., Holz, E., & Michael, T. (2014). Presence of a dog reduces subjective but not physiological stress responses to an analog trauma. Frontiers in psychology, 5, 1010.

Author Response

Thank for your comments and suggestions. We have made edits to the manuscript based on your feedback, described below in detail:

1. Methods section: The number of participants in the 4 different groups differ substantially. It is not clear to me how this happened. Different number of drop outs?  How were drop out defined? The authors should specify this and mention it in the Discussion section.

The slightly different group numbers in the groups was due to inherent challenges of random assignment with four conditions and a smaller sample. The study was limited due to having no funding, so data collection could not continue which would have resulted in larger and more equal group numbers. We have addressed this in the Discussion section under Limitations in lines 426-428, which reads: "Group size was uneven due to limitations to randomization to four groups with a smaller sample. However, because these differences were not due to bias or non-random factors, implications for validity of the study’s findings are likely minimal."

Also, thank you for pointing out that we had omitted information about dropout. Participants were able to voluntarily withdraw participation at any time, and protocols indicated that the stress procedure should be stopped if a participant experiences severe distress. This did not happen in the current study, but is now described in the Methods in lines 209-211.

2. Methods/Results: In line with this, an a priori sample size calculation is missing. Since there is abundant research on the effects of pet’s and social support on human stress responses, it should have be possible to estimate an effect size and perform a sample size calculation. The authors should at least report a post hoc power analysis for their non-significant findings.

In practicality, the sample size and data collection window was limited by funding rather than an a priori sample size estimation. Estimates of power were considered in the statistical approach to this data. However, we recognize that having a post-hoc power analysis presented in the paper would be beneficial and appreciate this recommendation. This information is now presented in lines 277-282, which reads: "A repeated measures, within-between interaction post hoc power analysis conducted with G*Power (version 3.1.9.7; Kiel University, Kiel, Germany) indicated that the sample of N = 132 and 4 groups was adequately powered at 0.93 to detect a medium effect (f = 0.15) with 2 repeated measurements (PA, NA, and NSE variables) and at 0.97 for 3 repeated measurements (SBP, DBP, and HR variables)."

4.       Methods/Discussion: Social evaluation in the cognitive stressor The authors state correctly that the TSST also has a cognitive component and might not have been exclusively social evaluative.  However, I did not really understand whether the cognitive stressor also had a social evaluative component? The authors write that: If a participant made a mistake, they were corrected and asked to continue from the correct number. Was this correction performed by the experimenter? Or automatically by a computer? If the correction was performed by the experimenter the two stressors both have cognitive and social-evaluative components integrated, which may have caused the failure to find an effect of the stressor type. It would be great if the authors could specify this in the Methods section and – if necessary – also integrate this point in the Discussion section.

We thank the reviewer for pointing this out. Indeed, mistakes were corrected by the researcher which is now more clear in line 216 of the Methods. We also added more discussion on this in lines 411-414, which reads: "In addition, the cognitive stressor contained a very small social component given that the experimenter was required to correct the participant in response to an error. This overlap in the nature of the stressors may explain why an effect of stressor type was not found. "

5.       Discussion section:  There are at least some other studies that I am aware of, which also did not find an effect of social support from a therapy dog on physiological stress indices, but only on subjective parameters (PANAS, STAI-S).  It might be helpful to include these in the Discussion section.

Great point, and thank you for this suggestion. We have added more discussion of the previous literature on this topic in lines 390-395, which reads: "Indeed, several studies have found significant stress-buffering effects of dog presence via reduced heart rate [19, 23], electrodermal activity [21], cortisol [27], and blood pressure [22] in response to a stressor compared to when alone or with a supportive human relationship. However, other studies have found that dog presence only impacts self-report measures (e.g., positive affect, state anxiety), but not physiological measures [53-55]."

This manuscript is a resubmission of an earlier submission. The following is a list of the peer review reports and author responses from that submission.

Round 1

Reviewer 1 Report

Comments and Suggestions for Authors

The Effect of Mental Activation of One’s Pet Dog on Stress Reactivity

Manuscript ID: ijerph-2432302

Summary

The manuscript describes a study that aimed to determine whether mental activation through thinking and writing about the pet reduces the stress response.

General concept comments

Article

The manuscript is very well written and synthesised, making it very easy to read and understand. It is understood that it is a relatively new topic, however, references are very old. A further literature review is suggested.

Review

There is much work to be done, as the results are not entirely robust. It is acknowledged that they accept their own limitations, but there are too many of them. Specific comments are listed below.

Specific comments

162-163. I suggest that the "heart rate and blood pressure" symbol as well as the "Self-reported affect" symbol go below table 1.

Reviewer 2 Report

Comments and Suggestions for Authors

Overall impression: Fascinating study, makes a unique contribution and extends the literature pertaining to animal-derived social support for humans.  

Introduction:  Good succinct summarization of literature related to current state of empirical research in this area. 

Conceptually, there needs to be more development to better contextualize/situate this work and maximize the important contribution it makes.  Social support in the literature is not a monolithic construct.  Rather, it has sub-dimensions such as instrumental (helping with a concrete task or issue such as giving someone a ride or picking up groceries) and emotional support (comfort, affection, etc.).  Animals can provide both for humans, e.g., they can learn to perform tasks to provide support (instrumental), and, they provide emotional support.  Historically, animals have been recognized as providers of instrumental social support (herding, guarding, hunting, service animals, etc.).  Animals are also increasingly being recognized for the emotional social support they provide to people, both in formal roles (e.g., therapy animals) and as companion animals and Emotional Support Animals.  Content needs to be added to briefly address the subdimensions of social support and historical/current ways animals have contributed to such for people. Creating a section such as "Background and Significance" (with the lit review and conceptual framing) and separating it from the "Introduction" would substantively strengthen this manuscript.

Research Question: Aims and hypotheses are clearly stated and emerge from identified gaps in the empirical lit review.

Design:  The design and analytic plan used are appropriate for the research aims and hypotheses.  Methods for data collection and analysis are clearly described.  Approval by the institution's IRB is clearly stated.  Sampling is adequate given the exploratory nature of the study.

Within the procedures, it is noted that participants were generalized to either the pet dog schema condition or the general positive memory condition.  It is not clear why a general positive memory was used as the comparison condition, rather than using no schema priming or a neutral schema such as a casual acquaintance as described in previous related research (lines 74-76) in the introduction section.  This rationale for this decision needs to be noted, and any subsequent impact it could have potentially had in findings needs to be addressed in the "Discussion" section.

Findings: Data analyses reported were stated with the purpose, design, and data collected.  The interpretations of the results were consistent with the raw data presented in the tables (which were very well-done).

Discussion: Overall well-done; authors carefully linked findings back to the limited body of related empirical literature.  The authors' point about the potential variability of one's views of their respective dogs is an important one, as it may have obfuscated significant differences in outcomes for those who actually do view their dog as a source of social support (as compared to those assigned to the general positive memory condition).  

As mentioned previously, using a general positive memory schema condition rather than a neutral schema or no schema needs to be addressed in light of the findings.  E.g., how may this procedure choice have potentially impacted findings related to cardiovascular reactivity, as compared to if a neutral or no schema had been used?  

his study has made a unique contribution, relating to  mental schemas of companion animals as a mitigator for negative affect during stress for this particular sample.  Need to bring it back to existing knowledge conceptually as well as empirically; in considering the two types of social support (emotional and instrumental) humans receive from animals - typically through their physical presence, this study offers some interesting findings!!

Author Response

please see attachement

Reviewer 3 Report

Comments and Suggestions for Authors

The paper aimed to determine whether the mental activation of one’s pet dog reduces stress reactivity from a subsequent experimental stressor. To do this, the study used 132 dog-owning participants randomly assigned to one of two mental activation conditions (pet dog; general) and one of two stressor conditions (social-evaluative; cognitive). Results suggest that mental activation of a pet dog may not be sufficient to elicit a social support schema to significantly impact cardiovascular stress reactivity. Overall, the paper is well written, and results will be of interest to the field industry. However, there are several limitations specifically with the sampling method including demographic biases, self-selection and potential impacts pre-experiment instructions on outcome. These limitations are well described in the discussion, although the section could be expanded to highlight their implications on the results and limitations for generalisation to a wider population. The paper could also be enhanced by expanding on the potential need for future research in the area.
